# A Suggested Modification to FIGO Stage IV Epithelial Ovarian Cancer

**DOI:** 10.3390/cancers15030706

**Published:** 2023-01-24

**Authors:** Marie Métairie, Louise Benoit, Meriem Koual, Enrica Bentivegna, Henri Wohrer, Pierre-Adrien Bolze, Yohan Kerbage, Emilie Raimond, Cherif Akladios, Xavier Carcopino, Geoffroy Canlorbe, Jennifer Uzan, Vincent Lavoué, Camille Mimoun, Cyrille Huchon, Martin Koskas, Hélène Costaz, François Margueritte, Yohann Dabi, Cyril Touboul, Sofiane Bendifallah, Lobna Ouldamer, Nicolas Delanoy, Huyen-Thu Nguyen-Xuan, Anne-Sophie Bats, Henri Azaïs

**Affiliations:** 1AP-HP (Assistance Publique des Hôpitaux de Paris), Department of Gynaecological Oncological and Breast Surgery, Hôpital Européen Georges-Pompidou, 75015 Paris, France; 2INSERM UMR-S 1124, University of Paris Cité, Centre Universitaire des Saints-Pères, 75006 Paris, France; 3Department of Gynaecologic and Oncologic Surgery and Obstetrics, Lyon Sud University Hospital, Hospices Civils de Lyon, Université Lyon 1, 69002 Lyon, France; 4CHU Lille, Department of Gynaecologic Surgery, University Lille, 59000 Lille, France; 5Department of Obstetrics and Gynaecology, Institute Alix de Champagne University Hospital, 51100 Reims, France; 6Department of Gynaecology, Hôpitaux Universitaires de Strasbourg, 67200 Strasbourg, France; 7Department of Obstetrics and Gynaecology, Hôpital Nord, APHM, Aix-Marseille University (AMU), University Avignon, CNRS, IRD, IMBE, UMR 7263, 13397 Marseille, France; 8AP-HP (Assistance Publique des Hôpitaux de Paris), Department of Gynaecological and Breast Surgery and Oncology, Pitié-Salpêtrière, 75013 Paris, France; 9Centre de Recherche Saint-Antoine (CRSA), INSERM UMR_S_938, Cancer Biology and Therapeutics, Sorbonne University, 75012 Paris, France; 10University Institute of Cancer, Sorbonne University, 75013 Paris, France; 11Department of Obstetrics Gynecology and Reproductive Medicine, University Paris Est Créteil, Centre Hospitalier Intercommunal de Créteil, 94000 Créteil, France; 12Department of Gynaecological Surgery, INSERM U1085, équipe 8, CRLC Eugène Marquis, Université de Rennes 1, Hôpital Sud, CHU de Rennes, 35000 Rennes, France; 13Department of Gynaecological Oncological and Breast Surgery—Université de Paris, Hôpital Lariboisière, 75010 Paris, France; 14AP-HP (Assistance Publique des Hôpitaux de Paris), Division of Gynaecologic Oncology, Bichat University Hospital, 75018 Paris, France; 15Department of Surgical Oncology, Georges-François Leclerc Centre, 21000 Dijon, France; 16Department of Gynaecology, Centre Hospitalier Intercommunal de Poissy-Saint-Germain-en-laye, Site Hospitalier de Poissy, 78498 Poissy, France; 17AP-HP (Assistance Publique des Hôpitaux de Paris), Department of Gynaecology ans Obstetrics, Hôpital Tenon, 75020 Paris, France; 18Department of Gynaecology, Hôpital Universitaire de Tours, 37000 Tours, France; 19AP-HP (Assistance Publique des Hôpitaux de Paris), Department of Medical Oncology, Hôpital Européen Georges-Pompidou, 75015 Paris, France; 20INSERM UMR-S 1147, University of Paris Cité, Centre de Recherche des Cordeliers, 75006 Paris, France; 21Institut du Cancer Paris CARPEM, 75006 Paris, France

**Keywords:** epithelial ovarian cancer, FIGO stage IV, metastatic patterns, prognosis, pleural involvement, overall survival

## Abstract

**Simple Summary:**

The identification of prognostic factors is important to improve the management of patients with ovarian cancer (OC). The staging classification of OC was revised in 2014 and 2018 by the FIGO (International Federation of Gynecology and Obstetrics) Gynecological Oncology Committee and dichotomizes stages IV into stage IVA and IVB. The FIGO classification aims to establish a disease severity scale and to group patients with similar prognoses. The objective of our retrospective, multicenter study was to assess the prognostic impact of this dichotomization and of the initial metastatic localization of patients with FIGO stage IV OC. We showed that, among our 307 patients, FIGO stage IVA patients had a worse prognosis than FIGO stage IVB patients. The initial pleural effusion was a factor of poor prognosis in terms of overall survival. We suggest a modification of the current FIGO staging classification.

**Abstract:**

International Federation of Gynecology and Obstetrics (FIGO) staging classification for stage IV epithelial ovarian cancer (EOC) separates stages IVA (pleural effusion) and IVB (parenchymal and/or extra-abdominal lymph node metastases). We aimed to evaluate its prognostic impact and to compare survival according to the initial metastatic location. We conducted a multicenter study between 2000 and 2020, including patients with a FIGO stage IV EOC. Primary endpoint was overall survival (OS). The secondary endpoints were progression-free survival (PFS) and recurrence rates. We included 307 patients: 98 (32%) had FIGO stage IVA and 209 (68%) had FIGO stage IVB. The median OS and PFS of stage IVA patients were significantly lower than those of stage IVB patients (31 versus 45 months (*p* = 0.02) and 18 versus 25 months (*p* = 0.01), respectively). Recurrence rate was higher in stage IVA than IVB patients (65% versus 47% (*p* = 0.004)). Initial pleural involvement was a poor prognostic factor with a median OS of 35 months versus 49 months for patients without initial pleural involvement (*p* = 0.024). Patients with FIGO stage IVA had a worse prognosis than patients with FIGO stage IVB EOC. Pleural involvement appears to be relevant for predicting survival. We suggest a modification of the current FIGO staging classification.

## 1. Introduction

The purpose of the cancer staging system is twofold. It allows one to describe the tumor spread and to define groups of patients with identical prognoses. The staging of epithelial ovarian cancer (EOC) was revised in 2014 and 2018 by the International Federation of Gynecology and Obstetrics (FIGO) [1,2]. Stage IV was a single entity in 1988 and was separated by the new FIGO classification. Patients with cytologically proven pleural effusion and/or metastasis are classified as stage IVA. Those with extra-abdominal parenchymal and/or lymph node metastasis are classified as stage IVB. Patients with inguinal lymph node metastasis or transmural digestive involvement with mucosal involvement, previously classified as stage III, became stage IVB.

The aim of the FIGO classification is to group together patients with similar prognoses. The prognostic value of the subdivision into FIGO stages IVA and IVB remains controversial. The available data on the prognostic impact of initial metastatic locations are few and inconsistent [3,4]. Several studies have shown no significant difference in overall survival (OS) between FIGO stages IVA and IVB [5,6,7,8]. In contrast, in 2018, a study of 160 patients showed that patients with stage IVB tumors had better OS than stage IVA patients and benefited more from neoadjuvant chemotherapy [9]. In 2018, a study of 551 patients with serous-type OC showed better OS for patients with extra-abdominal lymph node involvement alone with median OS of 41.4 months versus 25.2 and 26.8 months, respectively, for patients with metastatic pleural effusion and other/multiple distant metastatic sites [10].

The aim of our study was to evaluate the prognostic impact of the separation into stages IVA and IVB and of the initial metastatic location, especially the initial pleural involvement, of patients with stage IV EOC.

## 2. Materials and Methods

We conducted a retrospective cohort study of patients with EOC. The data were issued from a French multicenter database composed of thirteen hospital centers (FRANCOGYN research group). This database included patients with EOC of any FIGO stage between July 2000 and December 2020. The study was approved by the Institutional Ethics Committee of the National College of French Gynecologists and Obstetricians (CNGOF) (CEROG #2022-GYN-0803) [11].

### 2.1. Inclusion Criteria

Included patients had stage IV EOC according to the 2018 FIGO classification of ovarian cancer [2]. Given the evolution of the FIGO classification during the implementation of the cohort, all patients’ tumors were restaged according to the latest FIGO 2018 classification. Patients with tumor pleural effusion or pleural metastasis were classified as stage IVA. Pleural cytology by fine needle or thoracoscopy was performed in patients with pleural effusion to confirm the presence of malignant cells. Patients with parenchymal metastasis, including transmural bowel, or lymph node, supra-diaphragmatic or inguinal, were classified as stage IVB. Patients with pleural tumor effusion and/or pleural metastasis (FIGO IVA) and supra-diaphragmatic and/or inguinal lymph node involvement and/or parenchymal metastasis (FIGO IVB) were considered stage IVB patients. Initial metastatic locations were determined from diagnostic imaging (computed tomography (CT) with or without PET-CT) and/or from pathological examination of biopsy or pleural cytology [12,13].

### 2.2. Exclusion Criteria

Patients with a non-epithelial ovarian tumor were excluded from the study [14]. Patients with undetermined FIGO stage were excluded from the study.

### 2.3. Care Management

Socio-demographic, clinical, biological and imaging data were collected for all patients. The diagnosis of EOC was based on pathological examination from a radio-guided or a surgical biopsy. An initial laparoscopy evaluation was used to define the peritoneal carcinomatosis index (PCI) and the Fagotti score [15,16]. Three treatment regimens were discussed, according to the current recommendations: primary cytoreductive surgery (PCS), neo-adjuvant chemotherapy with interval cytoreductive surgery (NACT-IS) and chemotherapy alone (CTA) [17,18,19,20]. The surgical goal was the absence of gross residual disease [21,22]. Residual disease was defined at the end of the intervention and separated in complete resection, residual disease < 10 mm and residual disease ≥ 10 mm [23]. Postoperative complications were assessed according to the Clavien–Dindo classification [24,25]. This classification allows postoperative complications to be graded on a scale from 0 (no complication) to 5 (death). Stage 1 and 2 complications were those requiring medical treatment. Severe complications required surgical, radiological or endoscopic treatment (stage 3) or were life threatening (stage 4).

### 2.4. Outcomes

We compared survival between stage IVA and IVB patients. The primary outcome was OS. The secondary outcomes were progression-free survival (PFS) and recurrence rates.

### 2.5. Statistical Analysis

Not always having proof that the death was caused by EOC, OS was defined as the time from the date of initial diagnosis to death from any cause. PFS was defined as the time from initial diagnosis to tumor recurrence. In the absence of recurrence, it was censored at the date of last news or death. Continuous variables were expressed using the mean ± standard deviation or using the median and the minimum and maximum values. They were compared using the Student’s *t*-test or the Wilcoxon test in the case of a non-parametric distribution. Categorical values were expressed using an absolute number and a percentage. They were compared using a Chi^2^ test or a Fischer exact test. Kaplan–Meier curves were used to graphically express the differences in OS and PFS. Comparison of the curves was performed by a log-rank test. All statistical analyses were performed using an Excel database (Microsoft, Redmond, WA, USA) and RStudio software version 2022.02.0.

## 3. Results

### 3.1. Descriptive Analysis of the Population

Between 2000 and 2020, 2252 patients from the FRANCOGYN cohort were managed for an EOC, all FIGO stages combined. Of these, 307 (13.6%) were included in our study: 98 (31.9%) in the FIGO IVA group and 209 (68.1%) in the FIGO IVB group. A flow chart of the study is shown in Figure 1.

The main patient characteristics are detailed in Table 1. FIGO IVA patients were significantly older than FIGO IVB patients (*p* = 0.03) and therefore significantly more postmenopausal (*p* < 0.005). The American Society of Anesthesiologists (ASA) score was used to assess the preoperative health status of patients. Patients were assigned a score ranging from 1 to 6 if they were in good general condition (ASA 1), had moderate (ASA 2), severe (ASA 3) or life-threatening (ASA 4) organ dysfunction, with a life expectancy of less than 24 h (ASA 5), or in a state of encephalic death (ASA 6). The mode of discovery of EOC was similar for stage IVA and stage IVB patients. Due to the retrospective nature of our study, the data concerning the tumor histology of 13 patients (6 patients with FIGO stage IVA and 7 patients with FIGO stage IVB) could not be specified beyond epithelial ovarian adenocarcinoma. More than half of the stage IVB patients had concomitant pleural involvement. Of the 209 stage IVB patients, 102 had parenchymal metastasis, 73 (36.1%) patients had intrahepatic metastasis and 43 (21.3%) patients had extrahepatic parenchymal involvement (brain, pancreatic, splenic, transmural involvement of the digestive tract). Almost 70% of stage IVB patients had lymph node involvement, mostly above the diaphragm (64.9%).

### 3.2. Descriptive Analysis of the Treatment Regimen

Of the 307 patients in our cohort, 76 (25%) were treated with CTA, 48 (16%) with PCS and 179 (59%) with NACT-IS. Of the patients with FIGO stage IVB EOC, 4 (1.9%) received no treatment. The treatment regimen did not differ between the two groups (*p* = 0.07). Platinum sensitivity in patients with stage IVB tumors tended to be lower than in stage IVA patients. Of the 98 stage IVA patients, 62 (65.3%) had a recurrence during the study period compared to 99 (47.4%) of the stage IVB patients (*p* = 0.004). The number of recurrences was identical regardless of FIGO stage, IVA or IVB. Median OS was 31 months for stage IVA patients versus 48 months for stage IVB patients (*p* = 0.02). Median PFS was 18 months for stage IVA patients versus 25 months for stage IVB patients (*p* = 0.01). The characteristics of the treatment regimens received by patients according to FIGO stage are detailed in Table 2.

### 3.3. Descriptive Analysis of Surgery for Operated Patients

Of the 307 patients in our cohort, 227 (73.9%) underwent surgery: 78 (80%) stage IVA patients and 149 (71%) stage IVB patients. The initial PCI was significantly higher in stage IVA patients (*p* = 0.03). There were significantly more bowel resections (*p* < 0.001) and more lymph node dissections (*p* = 0.01) in stage IVB patients. Supra-mesocolic surgery included splenectomy, cholecystectomy, liver resection and diaphragmatic peritoneum resection. The rate was similar in both groups. There were significantly more intraoperative complications in stage IVB patients (*p* = 0.01). There were more stage IVA patients with postoperative macroscopic tumor residue (*p* = 0.03). There were more severe postoperative complications, Clavien–Dindo grade ≥ 3, in the FIGO IVB group compared to the FIGO IVA group (17.4% versus 8.2%, *p* = 0.05). These results are detailed in Table 3.

### 3.4. Survival Analysis by FIGO Stage

Patients with stage IVA EOC had poorer 5-year OS and PFS than patients with stage IVB EOC, with 20% versus 38% (*p* = 0.02) and 13% versus 31%, respectively (*p* = 0.01). The median OS and PFS of stage IVA EOC patients compared to stage IVB patients were 31 months versus 48 months and 18 months versus 25 months, respectively. Kaplan–Meier survival curves according to initial FIGO stage are shown in Figure 2.

### 3.5. Survival Analysis by Initial Metastatic Location

Patients with an initial pleural involvement, FIGO stages IVA and IVB combined, had poorer median OS compared to patients without initial pleural involvement, 35 months versus 49 months, respectively (*p* = 0.024). These patients also had poorer 5-year OS compared to patients without initial pleural involvement, 29% versus 43%, respectively (*p* = 0.024). The initial pleural involvement did not significantly affect the PFS of patients with stage IV EOC. Kaplan–Meier survival curves according to initial pleural involvement are shown in Figure 3.

Patients with only initial node metastasis, supra-diaphragmatic and/or inguinal, had a median OS of 55 months and a 5-year OS of 48%. The difference in 5-year OS according to initial metastatic location was significant (*p* = 0.015). The initial metastatic location (lymph node only, pleural or parenchymal or mixed parenchymal and lymph node) did not significantly affect the PFS of patients with stage IV EOC. Kaplan–Meier survival curves according to initial metastatic location are shown in Figure 4.

## 4. Discussion

This study showed that patients with stage IVA EOC had poorer median and 5-year OS and poorer median and 5-year PFS than patients with stage IVB EOC. FIGO stage IVA patients had more recurrences than FIGO stage IVB. More specifically, initial pleural involvement beyond stage IVA or IVB was a poor prognostic factor with a 5-year OS of 29% and a median OS of 35 months.

These results are consistent with those of Tajik et al., who analyzed the prognostic impact of this dichotomy in 160 patients with stage IV EOC. However, they described a positive impact of NACT in patients with stage IVB EOC that we did not find [9]. Some studies have not shown an impact of this separation in terms of survival. Rosendahl et al. found a similar 5-year OS for the 149 stage IVA and 613 stage IVB patients, of 13% in both groups, which was well below our results [6]. Paik et al. studied the survival of 94 patients and showed no prognostic impact of this dichotomy [26]. Neither of these two studies specified the treatment regimens received by the patients. Two other studies found no prognostic impact of separating EOCs into FIGO IVA and IVB but only studied patients treated by PCS [7,27].

Several factors may have contributed to the difference in prognosis between stage IVA and stage IVB patients. Since 2014, EOCs with transmural digestive involvement are classified as FIGO stage IVB. In our cohort, 5% of patients had transmural digestive involvement and were therefore classified as stage IVB. This frequency was consistent with that found in the literature [7]. However, in 2019, Mert et al. found no prognostic significance of the depth of invasion of the rectosigmoid wall in 85 patients with stage IIIC and IV EOC [28]. In 2016, Ataseven et al. showed that the prognosis of patients classified as stage IVB with only a digestive metastasis was superior to that of other stage IV patients. This may have contributed to the difference in OS between stage IVA and IVB [29]. However, this was a rare event and its impact was probably relative. The 33 patients in our cohort classified as stage IVB only on node metastasis, inguinal or supra-diaphragmatic, had a 5-year OS of 48% and a median OS of 55 months. These results were better than the 5-year and median OS of the overall stage IVB patient population, 38% and 48 months, respectively. These results are consistent with those of the Tajik et al. study, which found a 5-year OS of approximately 40% versus less than 20% for other stage IVB patients (*p* = 0.04). However, this study only included a sub-population of 15 patients [9]. The study of 151 patients classified as FIGO stage IVB only on inguinal metastasis showed no difference in 5-year OS compared to 4403 patients with pelvic and/or para-aortic lymph node involvement, 46.3% versus 44.9%, respectively (*p* = 0.4). Their 5-year OS was better than that of the other 5956 stage IVB patients (*p* < 0.001) [30]. These results highlight the heterogeneity of patients grouped in FIGO stage IVB and help explain the difference in OS between stages IVA and IVB.

The 124 patients in our cohort with pleural effusion and lymph node metastasis and/or parenchymal metastasis were classified as stage IVB. However, as stage IVA patients have a worse prognosis than stage IVB patients, this choice is debatable. Nevertheless, it only underestimated the difference in OS between the two groups. Furthermore, stage IVB patients had more digestive resection and lymphadenectomy, whereas FIGO stage IVA patients had a higher mean initial PCI than FIGO stage IVB patients, 20 versus 15, respectively. This raises the question of the tumor profiles of stage IV EOCs. It can be hypothesized that the pleural effusion is merely a continuation of the intraperitoneal disease beyond the diaphragmatic domes and reflects the extent of the intraperitoneal disease. Finally, the postoperative tumor residue of stage IVA patients was higher than that of stage IVB patients, 50% and 25% residual disease, respectively. Residual disease is a widely identified prognostic factor and probably contributed to this difference in overall survival between the two groups. However, we demonstrated the impact of initial pleural effusion by pooling FIGO stage IVA and FIGO stage IVB patients and showing a significant difference in overall survival (*p* = 0.015). The pool of IVA and IVB patients with or without pleural involvement shows the impact of pleural involvement beyond complete resection. This is potentiated by the fact that pleural involvement is associated with diffuse peritoneal involvement and therefore with fewer complete resections. In addition, Perri et al. have shown that recurrences in the thorax alone are rare and emphasize the importance of controlling abdominal disease [31]. This provides further support for the theory that pleural effusion is a reflection of the extent of intraperitoneal disease. Based on these considerations, we suggest a change in the staging of stage IV EOC, considering initial pleural location.

## 5. Conclusions

Patients with FIGO stage IVA EOC had a significantly worse prognosis than patients with FIGO stage IVB EOC. Pleural involvement status at diagnosis appears to be relevant for predicting survival in stage IV EOC. As the FIGO classification aims to establish a disease severity scale and to group patients with similar prognoses, we propose to define a new FIGO stage IV classification of EOC. We propose to group patients with extra-abdominal lymph node involvement and/or parenchymal metastasis into FIGO stage IVA and to classify all patients with initial pleural involvement into FIGO stage IVB.

## Figures and Tables

**Figure 1 cancers-15-00706-f001:**
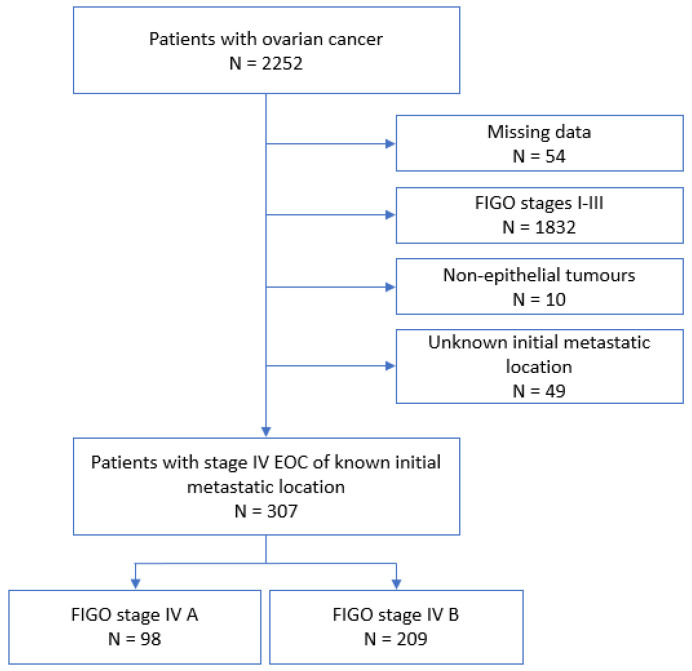
Flow chart. EOC: epithelial ovarian cancer.

**Figure 2 cancers-15-00706-f002:**
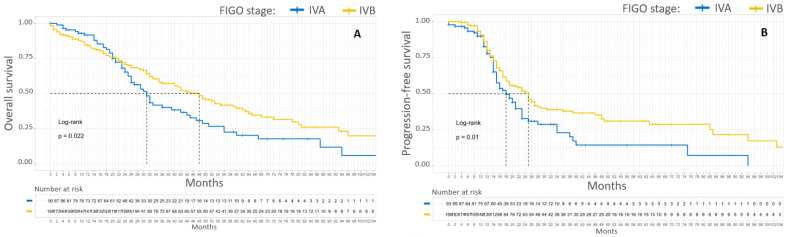
Kaplan–Meier OS (**A**) and PFS (**B**) curves for patients with FIGO stage IV EOC. In blue, patients with stage IVA EOC. In yellow, patients with stage IVB EOC.

**Figure 3 cancers-15-00706-f003:**
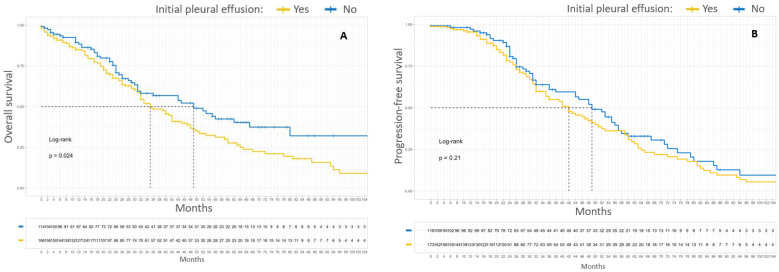
Kaplan–Meier OS (**A**) and PFS (**B**) curves for patients with FIGO stage IV EOC according to initial pleural effusion. In yellow, patients with pleural initial effusion. In blue, patients without initial pleural effusion.

**Figure 4 cancers-15-00706-f004:**
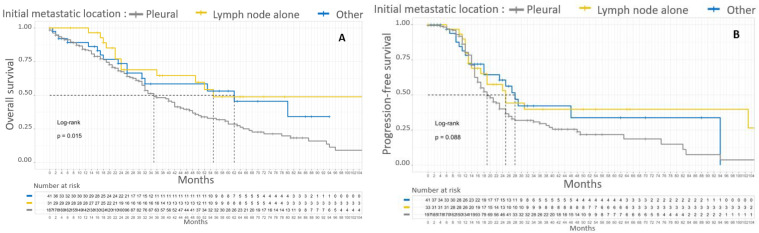
Kaplan–Meier OS (**A**) and PFS (**B**) curves for patients with FIGO stage IV EOC according to initial metastatic location. In gray, patients with pleural initial location. In yellow, patients with lymph node involvement alone. In blue, patients without pleural effusion with parenchymal metastasis with or without lymph node involvement.

**Table 1 cancers-15-00706-t001:** Characteristics of patients and their tumors according to FIGO stage. SD: standard deviation, HRT: hormone replacement therapy, ASA score: American Society of Anesthesiologists score, ASA 1: good general condition, ASA 2: moderate organ dysfunction, ASA 3: severe organ dysfunction, ASA 4: life-threatening organ dysfunction, ASA 5: life expectancy of less than 24 h, ^a^: abdominopelvic pain, increased abdominal circumference, transit disorder, metrorrhagia, ^b^: adenopathy, dyspnea, altered general condition, ^c^: carcinosarcoma, mixed, undifferentiated, Brenner’s tumor, ^d^: pancreas, spleen, digestive mucosa, brain.

	FIGO Stage IVA N = 98	FIGO Stage IVB N = 209	*p*-Value
Age at diagnosis (years) Mean ± SD Median (rank)			0.03
	65.2 ± 9.9	62.3 ± 11.9	
	66.0 (41.0–86.0)	62.0 (34.0–89.0)	
Menopause N (%) HRT N (%)	93/95 (97.9)	171/202 (84.7)	<0.005
	14/71 (19.7)	25/135 (18.5)	0.98
ASA score N (%) 1			0.76
	24 (24.5)	62 (29.7)	
2 3	25 (25.5)	53 (25.4)	
	18 (18.4)	31 (14.8)	
4	1 (1.0)	2 (1.0)	
NA	30 (30.6)	61 (29.1)	
Mutation N (%)			1
BRCA1	7 (7.1)	12 (5.7)	
BRCA2	3 (3.1)	13 (6.2)	
Not sought	54 (55.1)	95 (45.5)	
Non-mutated	34 (34.7)	89 (42.6)	
Personal history N (%)			
Pelvic cancer	3/64 (4.7)	3/189 (1.6)	0.35
Breast cancer	2/67 (3.0)	10/196 (5.1)	0.71
Hysterectomy	3/67 (4.5)	12/194 (6.2)	0.83
Ovarian surgery	3/64 (4.7)	7/188 (3.7)	1
Family history N (%)			
Gynaecological cancer	21/79 (26.6)	64/182 (35.2)	0.22
Other cancers	10/60 (16.7)	47/175 (26.9)	0.16
Discovery mode N (%)			
Imagery	3/72 (4.2)	12/136 (8.8)	0.34
Abdominal symptomatology ^a^	43/72 (59.7)	78/136 (57.4)	0.86
Extra-abdominal symptomatology ^b^	41/72 (56.9)	69/136 (50.7)	0.48
Histology N (%)			0.57
Serous	79 (80.6)	169 (80.9)	
High grade	59/65 (90.8)	133/141 (94.3)	
Low grade	6/65 (9.2)	8/141 (5.7)	
Mucinous	3 (3.1)	1 (0.5)	
Endometrioid	3 (3.1)	10 (4.8)	
Clear cells	2 (2.0)	6 (2.9)	
Other ^c^	5 (5.1)	16 (7.6)	
NA	6 (6.1)	7 (3.3)	
Initial metastatic site N (%)			<0.001
Pleural	98 (100)	124 (59.3)	0.04
Parenchymal	0 (0)	102/202 (50.5)	<0.001
Intra-hepatic	0 (0)	73/202 (36.1)	<0.001
Other ^d^	0 (0)	43/202 (21.3)	<0.001
Lymph node	0 (0)	138/202 (68.3)	<0.001
Supra-diaphragmatic	0 (0)	131/202 (64.9)	<0.001
Inguinal	0 (0)	10/202 (5.0)	0.06
CA 125 at diagnosis (U/mL)			0.14
≤500 N (%)	19/96 (19.8)	58/203 (28.6)	
>500 N (%)	77/96 (80.2)	145/203 (71.4)	

**Table 2 cancers-15-00706-t002:** Characteristics of the therapeutic regimens according to FIGO stage IVA or IVB. CTA: chemotherapy alone, PCS: primary cytoreductive surgery, NACT-IS: neoadjuvant chemotherapy and interval surgery, ACT: adjuvant chemotherapy, SD: standard deviation.

	FIGO IVA N = 98	FIGO IVB N = 209	*p*-Value
Treatment regimen N (%) CTA PCS			0.07
	20 (20.4)	56 (26.8)	
	11 (11.2)	37/205 (17.7)	
NACT-IS	67 (68.4)	112/205 (53.9)	
Chemotherapy N (%)			
Neoadjuvant Number of NACT: Mean ± SD	87 (88.8)	112 (53.6)	0.10
	5.1 ± 1.9	4.9 ± 1.7	0.37
Adjuvant Number of ACT: Mean ± SD			1
	2.8 ± 2.2	3.4 ± 2.8	0.55
Platinum sensitivity			0.21
Low	27/68 (39.7)	71/138 (51.4)	
Intermediate	19/68 (27.9)	36/138 (26.1)	
High	22/68 (32.4)	31/138 (22.5)	
Bevacizumab N (%)	33/76 (43.4)	80/159 (50.3)	0.40
Recurrence N (%)	62/95 (65.3)	99/209 (47.4)	0.004
CA-125 at relapse (U/mL)			0.65
Mean ± SD	278.1 ± 692.5	217.6 ± 424.2	
Median (rank)	103 (5–4000)	87 (5–2864)	
Chemotherapy at relapse	45/95 (47.4)	86/87 (98.9)	0.60
Surgery at relapse	4/35 (11.4)	13/65 (19.1)	0.47
Number of relapse			0.96
Mean ± SD	0.86 ± 1.47	0.87 ± 1.4	
Median (rank)	0 (0–8)	0 (0–6)	
Overall survival (months)			
Mean ± SD	29.2 ± 22.0	33.9 ± 28.0	0.15
Median (rank)	31.0 (1–106)	48.0 (0–125)	0.02
Progression-free survival (months)			
Mean ± SD	18.5± 14.8	22.5 ± 22.3	0.66
Median (rank)	18 (0–94)	25.0 (0–115)	0.01

**Table 3 cancers-15-00706-t003:** Characteristics of surgery for operated patients according to FIGO stage IVA or IVB. ^a^: colectomy including posterior pelvectomy, ^b^: partial or total cystectomy, JJ stent, ^c^: vascular injury or bleeding complication requiring transfusion of red blood cells, ^d^: splenic, hepatic or biliary tract injury.

	FIGO IVA N = 78	FIGO IVB N = 149	*p*-Value
PCI Initial PCI Mean ± SD			
			0.03
	20.4 ± 10.7	15.4 ± 9.5	
Median (rank)	25.0 (0–31)	14.0 (0–32)	
Operating time (minutes)			
Mean ± SD Median (rank)	350 ± 177.2	401.1 ± 199.4	0.69
	330 (120–645)	345 (120–690)	
Digestive resection N (%)	7/74 (9.5)	48/143 (33.6)	<0.001
Supra-mesocolic surgery ^a^ N (%)			
Splenectomy	6/63 (9.5)	14/125 (11.2)	0.92
Cholecystectomy	2/58 (3.4)	5/119 (4.2)	1
Liver resection	1/53 (1.9)	9/119 (7.6)	0.35
Diaphragmatic dome resection	25/74 (33.8)	65/144 (45.1)	0.13
Lymphadenectomy N (%)	32/74 (43.2)	91/146 (62.3)	0.01
Urinary surgical procedure ^b^ N (%)	1/70 (1.4)	1/143 (0.7)	1
Intra-operative complication N (%)	14/71 (19.7)	53/138 (38.4)	0.01
Urinary or digestive	1/71 (1.4)	14/138 (10.1)	0.24
Vascular injury or hemorrhagic complication ^c^	14/71 (19.7)	38/138 (27.5)	1
Pleural injury	1/71 (1.4)	10/138 (7.2)	0.52
Other ^d^	2/71 (2.8)	4/138 (2.9)	0.80
Postoperative residual disease N (%)			0.03
Complete resection	36/72 (50)	108/142 (76.1)	
Residual disease < 10 mm	10/72 (13.9)	12/142 (8.5)	
Residual disease ≥ 10 mm	26/72 (36.1)	22/142 (15.4)	
Postoperative complication according to the Clavien–Dindo classification N (%)			0.05
0	58/73 (79.5)	91/138 (65.9)	
1	0/73 (0)	9/138 (6.5)	
2	9/73 (12.3)	14/138 (10.2)	
3A	4/73 (5.5)	8/138 (5.8)	
3B	1/73 (1.4)	9/138 (6.6)	
4A	0/73 (0)	4/138 (2.9)	
4B	0/73 (0)	2/138 (1.4)	
5	1/73 (1.4)	1/138 (0.7)	

## Data Availability

Data are available upon request.

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
