# Peer review of "A Suggested Modification to FIGO Stage IV Epithelial Ovarian Cancer"

_cancers, 2023, doi:10.3390/cancers15030706_

Round 1
Reviewer 1 Report
This article raises the question about the FIGO stages IVA and IVB of epithelial ovarian cancer based on their survival data between two stages. Their data seems to have an impact on FIGO classification system of epithelial ovarian cancer. However, there are several minor questions shown below.
1. In statistical analysis, “OS was defined as the time from the date of initial diagnosis to death from any cause.”. Why did authors choose “death from any cause” instead of disease specific death?
2. In table 1, explanation for ASA score is lacking.
3. In table 1, patients with NA for histology appeared. How were those cases histologically diagnosed?
4. In care management, cytoreduction score of the residual disease is introduced as grading from CC0 to CC3. However, residual disease was not presented according to this grading system in the manuscript and the table 3.
5. In care management, Clavien-Dindo classification for postoperative complication is presented. Short explanation for the classification would be appreciated.
Reviewer 2 Report
1. The initial PCI was higher in FIGO IVA group and the complete resection rate was also lower in FIGO IVA group. Is there possible due to the lower complete resection rate in stage IVA patients result in worse 5-year PFS and OS, not because of pleural involvement?
2. The digestive resection rate and lymphadenectomy rate are higher in stage IVB group, is it because of these aggressive surgical procedure result in better outcome?
Reviewer 3 Report
The study is well conducted, well organized, and structured. The content is interesting, and the results are presented accurately and clearly. The conclusion is straightforward and argues for reconsidering the FIGO classification of ovarian cancer.
Minor objection: In Figure 1. Epithelial ovarian cancers (EOC) are shown at the top of the flowchart, and non-epithelial tumors appear on the right part of the flowchart. This is a little confusing. I did not notice that review of the existing pathological reports was mentioned in the text. Did the authors find misdiagnosed non-epithelial tumors among the ovarian epithelial cancers included in the study?
